# ReFSQL: A Retrieval-Augmentation Framework for Text-to-SQL Generation

**Kun Zhang[1,2], Xiexiong Lin[3]\*, Yuanzhuo Wang[1,2,4]\*, Xin Zhang[3], Fei Sun[1,2],**
**Jianhe Cen[4], Xuhui Jiang[1,2], Hexiang Tan[1,2], Huawei Shen[1,2]**

[1]Data Intelligence System Research Center, Institute of Computing Technology,
Chinese Academy of Sciences; [2]School of Computer Science and Technology,
University of Chinese Academy of Sciences; [3]Ant Group. [4]Big Data Academy, Zhongke;
{zhangkun18z, wangyuanzhuo, sunfei, jiangxuhui19g,tanhexiang21s}@ict.ac.cn
{shenhuawei}@ict.ac.cn {xiexiong.lxx}@antgroup.com {evanzhangxin}@gmail.com

## Abstract

Text-to-SQL is the task that aims at translating natural language questions into SQL queries. Existing methods directly align the natural language with SQL Language and train one encoder-decoder-based model to fit all questions. However, they underestimate the inherent structural characteristics of SQL, as well as the gap between specific structure knowledge and general knowledge. This leads to structure errors in the generated SQL. To address the above challenges, we propose a retrieval-argument framework, namely ReFSQL. It contains two parts, structure-enhanced retriever and the generator. Structure-enhanced retriever is designed to identify samples with comparable specific knowledge in an unsupervised way. Subsequently, we incorporate the retrieved samples' SQL into the input, enabling the model to acquire prior knowledge of similar SQL grammar. To further bridge the gap between specific and general knowledge, we present a mahalanobis contrastive learning method, which facilitates the transfer of the sample toward the specific knowledge distribution constructed by the retrieved samples. Experimental results on five datasets verify the effectiveness of our approach in improving the accuracy and robustness of Text-to-SQL generation. Our framework has achieved improved performance when combined with many other backbone models (including the 11B flan-T5) and also achieved state-of-the-art performance when compared to existing methods that employ the fine-tuning approach.

## 1 Introduction

Relational databases, which serve as ubiquitous components within data management systems, offer a means of storing heterogeneous data types, encompassing text, integer, float, and various other formats. However, the proficient utilization of managing databases by ordinary users remains a challenge, primarily due to their limited proficiency in

---

*Corresponding author.

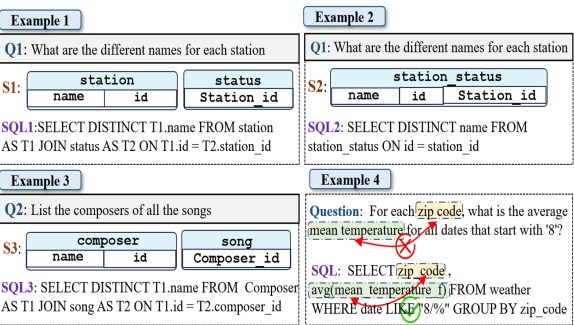

Figure 1: Example 1, 2, and 3 are three Text-to-Sql instances. Example 4 shows the difference in the grammar between natural language and SQL .

translating their information needs into the structured query language (SQL), which serves as the standard database language. To facilitate the querying process for non-professional users, researchers have introduced the Text-to-SQL task, which seeks to automate the translation of users' natural language questions into SQL queries.

Recently, with the rapid development of language models, many work (Qi et al., 2022; Li et al., 2023) have leveraged one encoder-decoder-based model to fit the entire training set, directly transformed SQL queries into a serialization structure and aligned them with natural languages. However, such methods can easily lead to structural errors in the generated SQL, e.g., missing "JOIN IN" operations, conducting incorrect comparison operations, etc. The observed phenomenon can be attributed to the underestimation of the inherent structural characteristics as well as the gap between specific knowledge and general knowledge.

Natural languages exhibit inherent characteristics where the appearance of each word is highly influenced by the preceding or following sequence. In contrast, as shown in Example 4 in Figure 1, SQL language possesses unique inherent structural characteristics, i.e., heavily relies on the database schema and SQL grammar but lacks robust natural language attributes. Previous methods that directly

align the distributions of natural language and SQL query may fail to capture these inherent structural characteristics in SQL .

In the real world, SQL structures exhibit significant variations due to diverse structural characteristics, such as different database schemas and complex SQL grammars. As illustrated in Figure 1, Example 1 and Example 2 undergo drastic changes in their SQL representations when the underlying schema differs. The conventional approach in Text-to-SQL is to train a one-size-fits-all model to fit the entire training set, acquiring the general knowledge, i.e., the distribution in the global space. However, such models struggle to acquire specific knowledge, i.e. the distribution of the local space, and adapt to the varied samples. The gap between specific and general knowledge significantly contributes to structural errors in the generated SQL. This discrepancy is particularly pronounced in cases with complex structures, where the general distribution learned by the model fails to adequately fit these cases unless a substantial amount of data, which can be prohibitively expensive in practice, is available.

To overcome these challenges, we draw inspiration from the process of human cognition. As depicted in Figure 1, humans can compose appropriate SQL queries by referring to similar samples that share common specific knowledge, including question semantics, schema structures, or SQL queries. In this paper, we propose a retrieval-augmented framework for Text-to-SQL generation. To identify samples with comparable specific knowledge, we design a structure-enhanced retriever that takes into account question semantics and schema structure. Subsequently, we incorporate the retrieved samples' SQL into the input, enabling our model to acquire prior knowledge of similar SQL grammar. To further enhance the model's ability to acquire specific knowledge, we employ a contrastive learning approach. As highlighted in (Kao et al., 2021), the utilization of contrastive learning offers a viable solution to mitigate the gaps arising from disparate distributions. The retrieved samples serve as building blocks for constructing the specific distribution that corresponds to the current sample. Through contrastive learning, we guide the samples toward the specific knowledge distribution. However, quantifying the distance from a sample to the distribution poses a challenge within contrastive learning. To address this, we introduce the

Mahalanobis distance as a measure of the distance between the sample and the specific distribution. By utilizing the Mahalanobis distance, we employ a contrastive loss function to facilitate the transfer of the sample toward the specific knowledge distribution.

In general, our main contributions are listed as follows:

- We propose a retrieval-augmentation framework for Text-to-SQL generation, which can adapt to samples with various inherent SQL characteristics and bridge the gap between specific knowledge and general knowledge.

- To further bridge the gap between specific and general knowledge, we present a mahalanobis contrastive learning method, which facilitates the transfer of the sample toward the specific knowledge distribution.

- We verify the effectiveness of the proposed framework on five widely-used datasets and achieve state-of-the-art performance in methods that employ the fine-tuning approach.

## 2 Preliminaries

Given a natural language query $\mathcal{Q}$ and a database schema $\mathcal{S} = (\mathcal{T}, \mathcal{C})$, the objective is to generate the corresponding SQL query $\mathcal{Y}$. The natural language query $\mathcal{Q}$ is represented as a sequence of tokens $\mathcal{Q} = \{q_i\}_{i=1}^{|\mathcal{Q}|}$, while the schema $\mathcal{S}$ consists of a collection of tables $\mathcal{T} = \{t_i\}_{i=1}^{|\mathcal{T}|}$ along with their associated columns $\mathcal{C} = \{\mathcal{C}_i\}_{i=1}^{|\mathcal{C}|}$. The content of the database $\mathcal{S}$ is denoted as $\mathcal{V}$. Each table $t_i$ contains a set of columns represented as $\mathcal{C}_i = \{c_{ij}\}_{j=1}^{|\mathcal{C}i|}$. Similarly, table names and column names are tokenized, such that a table name $t_i$ consists of $|t_i|$ tokens and the same applies to column names. In this study, the predicted SQL query is presented as a sequence of tokens, $\mathcal{Y} = \{y_i\}_{i=1}^{|\mathcal{Y}|}$.

## 3 Methodology

In this section, we will describe the proposed framework ReFSQL, which comprises two main components: the structure-enhanced retriever and the generator. The structure-enhanced retriever finds similar samples based on questions and schemas and constructing prompts using their corresponding SQL queries. On the other hand, the generator aims to bridge the gap between specific and general

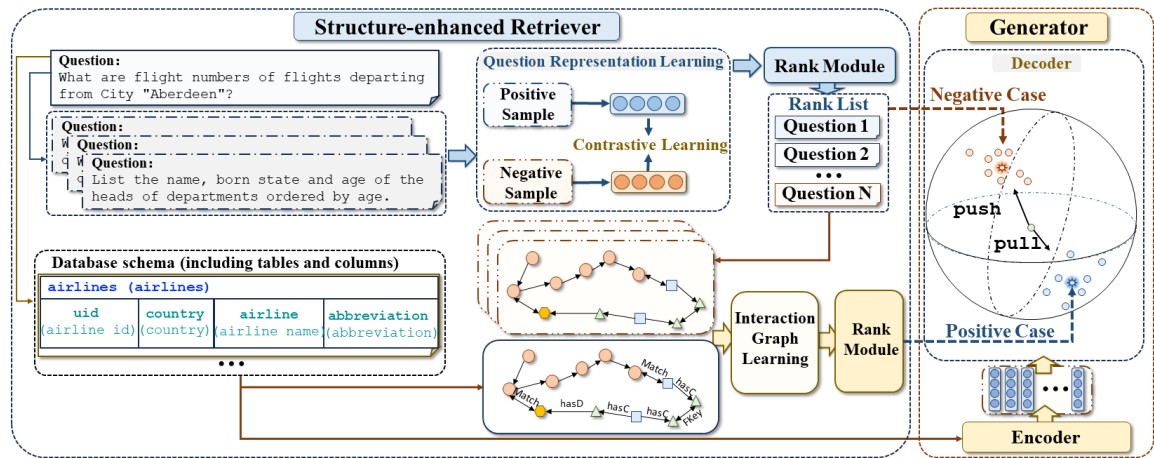

Figure 2: The overall architecture of our framework ReFSQL.

knowledge. It achieves this by employing the Mahalanobis distance to guide the sample toward the distribution of similar samples. Figure 2 provides an overview of the ReFSQL framework.

## 3.1 Structure-enhanced Retriever

In this section, we present the design of a structure-enhanced retriever aimed at identifying samples that exhibit a similar knowledge to the current sample. In Text-to-SQL tasks, the question and schema play crucial roles in generating accurate SQL queries. To enhance the representation of the question and schema, we leverage the question-SQL pairs and schema linking techniques. Thus, the structure-enhanced retriever consists of two modules: the linking-structure-enhanced schema retriever (LSES retriever) and the SQL-structure-enhanced question retriever (SQLSE retriever). We provide a detailed description of these modules below.

### 3.1.1 SQL-Structure-enhanced Question Retriever

In this section, we present an unsupervised method to enhance question representation via SQL structure information.

In our dataset, each sample comprises a SQL query and a corresponding question. In order to acquire effective question representations, we generate contrastive samples based on the similarity between SQL queries. Inspired by the methodology introduced in (Yu et al., 2018a), we leverage the tree structure of SQL queries to quantify their similarity. Specifically, we employ a straightforward yet powerful method called Tree-Edit-Distance-based (TED) similarity to measure the similarity

between two tree formats of SQL queries, denoted as $tr_1$ and $tr_2$. This similarity can be calculated using the following equation:

$$\delta_t = \text{TED}(tr_1, tr_2) \qquad (1)$$

Here, $\delta_t$ represents the similarity between $tr_1$ and $tr_2$. To construct the positive set, we randomly select $k$ samples with the highest similarity scores, while for the negative set, we also choose $k$ samples with the lowest similarity scores.

We employ the pre-trained Bert model (Devlin et al., 2018) with fixed parameters to encode the question and obtain its representation. The encoding process is performed as follows:

$$\mathbf{h}_{q_i} = \text{Bert}(q_i), \qquad (2)$$

Here, $\mathbf{h}_{q_i}$ denotes the representation of the question $q_i$.

We adopt the contrastive learning framework proposed in (Chen et al., 2020) and employ a cross-entropy objective with negative samples as described in (Chen et al., 2017). For each sample $q_i$, we denote the positive sample as $q_i^+$ and the negative sample as $q_i^-$. The training objective for the pair $\left(h_{q_i}, h_{q_i^+}\right)$ with $k$ pairs is defined as follows:

$$\mathcal{L}_{sc} = -\log\left(\frac{\exp(D_{cos}\left(h_{q_i}, h_{q_i^+}\right)/\tau)}{\sum_{q_j^* \in Q}\exp(D_{cos}\left(h_{q_i}, h_{q_i^*}\right)/\tau)}\right) \quad (3)$$

Here, $q_j^* \in Q$ represents the sample constructed from positive and negative samples. $\tau$ is a temperature hyperparameter, and $D_{cos}\left(\mathbf{h}_1, \mathbf{h}_2\right)$ denotes the cosine similarity $\frac{\mathbf{h}_1^\top \mathbf{h}_2}{|\mathbf{h}_1| \cdot |\mathbf{h}_2|}$.

After obtaining the learned question representations, we employ the cosine similarity function to rank the samples. Finally, we select the top $m$ similar samples for each sample.

### 3.1.2 Linking-Structure-based Schema Retriever

After we obtain $m$ samples by the previous Section 3.1.1, we utilize neural-based models to perform a reranking task with a focus on the schema structure.

In this section, we introduce the linking-structure-based schema retriever. The construction of the interaction graph, which captures the diverse relations between questions and databases, is elaborated. Following that, we adopt an unsupervised learning approach to derive the representation of the interaction graph. By leveraging this graph representation, we retrieve the relevant samples from the dataset.

**Interaction Graph Construction** We extract a diverse range of relations as triplets, establishing connections between tokens derived from both the question and the schema. These triplets are then seamlessly integrated to form a comprehensive graph representation. Our primary focus centers around two distinct types of relations: schema encoding and schema linking.

- Schema Encoding. Schema encoding relations pertain to the interconnections among schema items, explicitly delineating the structural information encapsulated within a database schema. Notable instances of such relations encompass BELONGS-TO, which denotes the association between a column and its corresponding table, and FOREIGN-KEY, which establishes a linkage between the foreign key in one table and the primary key in another table. By comprehensively capturing these schema encoding relations, a more holistic understanding of the schema's internal structure can be attained.

- Schema linking. Schema linking relations encompass the associations between schema and question items. In line with RAT-SQL (Wang et al., 2020), we leverage n-gram matches to signify instances wherein the question references specific schema items. However, the detection of these relations has proven to be a formidable task in prior investigations, primarily owing to the prevalent discrepancy between natural language expressions and the

explicit names assigned to schema components. Consequently, we employ a distinction between exact matches and partial matches to mitigate the deleterious impact of imprecise correspondences, thereby minimizing the noise engendered by imperfect matches.

**Interaction Graph Learning** The interaction graph learning module consists of two trainable components: the graph encoder and the edge decoder. The graph encoder encodes the interaction graph and the edge predictor is used as a downstream task. As mentioned above, the interaction graph is a heterogeneous graph. To model the content and structure of the interaction graph, we leverage R-GCN to encode the interaction graph. We denote the interaction graph of $i$th sample as $g_i$, then the graph embedding $\mathbf{h}_{g_i}$ is calculated as,

$$\mathbf{h}_{g_i} = \text{R-GCN}(g_i) \tag{4}$$

Then we use graph embedding learned above to retrieve the most similar samples through the cosine similarity function.

We design a simple yet effective SQL prompt to make the language model learn the SQL character during encoding. The input of $i$th sample mainly contains three parts, the question $\mathcal{Q}_i$, the database schema $S$, involving $\mathcal{S}_i = <\mathcal{T}, \mathcal{C}>$, and the SQL prompt $SP_i$. We follow (Qi et al., 2022) to serialize the inputs. Formally,

$$SP_i = \overline{\text{the similar SQL was:} sql_{i_1}|sql_{i_2}|\cdots, sql_{i_j},}$$
$$X_i = \overline{\mathcal{Q}_i|S_i|t_1 : c_{11}, \cdots, c_{1|T_1|} \mid t_2 : c_{21}, \cdots |SP_i} \tag{5}$$

where $sql_{i_j}$ is denoted as the $j$th SQL of similar samples for the $i$ sample. $t_i$ is the table name, $c_{ij}$ is the $j$-th column name of the $i$-th table. We adopt the "|" symbol to delineate the boundaries between $\mathcal{Q}$, $S$, different tables, and SQL prompts. We utilize the ":" symbol to differentiate between the table name and its corresponding columns.

### 3.2 Generator

In this section, we will describe the generator, which consists of two components: the encoder and the decoder.

### 3.2.1 Encoder

In our framework, encoders can be replaced, so we take the pre-trained model T5 as an example. For the $i$th sample,

$$\mathbf{h}_{X_i} = \text{T5-Encoder}(X_i) \tag{6}$$

where $\mathbf{h}_{X_i}$ is denoted as the encoded state of $X_i$.

### 3.2.2 Decoder

As described in Section 1, there exists a gap between the specific knowledge and the general knowledge which disturbs the learning of samples. To further bridge this gap, we introduce contrastive learning to guide the sample representation toward the distribution of similar samples and farther away from the distribution of dissimilar samples.

**Contrastive Sample Construction** To optimize the efficacy of the contrastive learning approach, we propose a more refined strategy for constructing contrastive samples. Following the methodology outlined in Section 3.1.1, we identify the top $m$ samples, denoted as set $M$. From this set, we select the top $k$ samples after applying the linking-structure-enhanced retriever for constructing positive samples. Conversely, we construct a negative sample set from the remaining lower-ranked samples in $M$. This process allows us to derive a distribution of similar semantic representations based on the positive samples, represented as $f_\phi(\mathbf{h}_{x^+}) \sim N(\mu^+, \sigma^{2+})$. Simultaneously, we obtain a distribution of dissimilar semantic representations based on the negative samples, denoted as $f\phi(\mathbf{h}_{x^-}) \sim N(\mu^-, \sigma^{2-})$.

**Mahalanobis Contrastive Learning**

To transfer the sample representation toward a distribution of similar semantic samples and distance it from a distribution of dissimilar samples, we propose the employment of the Mahalanobis contrastive mechanism (Li et al., 2022b). This mechanism aims to minimize the margin between the sample representation $\mathbf{h}_{x_i}$ and the similar semantic distribution $f_\phi(\mathbf{h}_{x^+}) \sim N(\mu^+, (\sigma^2)^+)$, while simultaneously maximizing the margin between the sample representation $\mathbf{h}_{x_i}$ and the dissimilar semantic distribution $f_\phi(\mathbf{h}_{x^-}) \sim N(\mu^-, (\sigma^2)^-)$.

The contrastive loss, denoted as $\mathcal{L}_{ma}$, can be defined as follows:

$$\mathcal{L}_{MA} = -\log\left( \frac{\exp\left(D_{ma}\left(f_\phi(\mathbf{h}_{x_j^+}), \mathbf{h}_{x_i}\right)/\tau\right)}{\sum_{x_j \in M} \exp\left(D_{ma}\left(f_\phi(\mathbf{h}_{x_j}), \mathbf{h}_{x_i}\right)/\tau\right)} \right) \tag{7}$$

Here, $x_j^* \in M$ represents the samples retrieved by the current sample $x_i$, and $D_{ma}$ refers to the Mahalanobis distance (De Maesschalck et al., 2000)

between the representation of the current sample $x_i$ and the distribution of retrieved samples $f_\phi(\mathbf{h}_{x_i}) \sim N(\mu, \sigma^2)$. This Mahalanobis distance can be calculated as $(\mathbf{h}_{x_i} - \mu)\sigma^2 I(h_{x_i} - \mu)$, where $I$ denotes the identity matrix. The hyperparameter $\tau$ controls the temperature of the contrastive loss.

### 3.2.3 Training Details

The loss in the decoding stage mainly consists of two parts, the MLE loss, and the contrastive loss. Specifically, the MLE loss, denoted by $\mathcal{L}_{CE}$, is computed as follows:

$$\mathcal{L}_{CE} = -\frac{1}{N} \sum_{i=1}^{N} \sum_{j=1}^{V} y_{i,j} \log \hat{y}_{i,j}, \tag{8}$$

Here, $N$ is the number of samples in the training data, while $V$ represents the vocabulary size. $y_{i,j}$ denotes the true label of the $j$-th word in the $i$-th sample, and $\hat{y}_{i,j}$ represents the corresponding predicted probability generated by the model.

Thus, the overall loss is computed as follows:

$$\mathcal{L} = \mathcal{L}_{CE} + \mathcal{L}_{MA}, \tag{9}$$

.

## 4 Experiment

In this section, we present a comprehensive evaluation of our proposed framework, ReFSQL, using five extensively utilized benchmark datasets. The obtained results serve as strong evidence to validate the effectiveness of our method.

### 4.1 Datasets and Preprocessing

We conduct experiments on several widely-used Text-to-SQL datasets. The details are shown below.

**Spider** Spider (Yu et al., 2018b) is a challenging benchmark that poses significant demands on the cross-domain and multi-table Text-to-SQL task. The dataset consists of a training set with 7,000 samples, a dev set with 1,034 samples, and a concealed test set with 2,147 samples.

**Spider-DK, Spider-Syn, and Spider-Realistic** To evaluate the robustness of our model, we employ a training strategy using the Spider training set while assessing its performance on three distinct evaluation sets: Spider-DK (Gan et al., 2021b), Spider-Syn (Gan et al., 2021a), and Spider-Realistic (Deng et al., 2020). The Spider-DK evaluation set consists of 535 samples and focuses on the integration of domain knowledge to rephrase

Table 1: EM and EX results on Spider's development set, and EM and F1 results on WikiSQL's test set (%).

| Approach | Spider | | WikiSQL | |
|---|---|---|---|---|
| | DevEM. | DevEX. | TestEM. | TestF1. |
| RAT-SQL+GAP+NatSQL (Gan et al., 2021c) | 73.7 | 75.0 | - | - |
| BRIDGE (Lin et al., 2020) | 71.1 | 70.3 | 85.7 | 91.1 |
| LGESQL + ELECTRA (Xu et al., 2020) | 75.1 | - | - | - |
| T5-3B(Shaw et al., 2020) | 71.5 | 74.4 | - | - |
| UnifiedSKG(T5-Base) (Xie et al., 2022) | 71.7 | - | 86.0 | - |
| UNISAR (Dou et al., 2022) | 70.0 | - | 86.7 | 91.7 |
| Uni-Parser(T5-Base) (Liu et al., 2022) | 61.2 | - | 85.8 | 91.3 |
| Uni-Parser(T5-Large) (Liu et al., 2022) | - | - | 86.9 | 92.1 |
| ChatGPT (Liu et al., 2023) | - | 70.1 | - | - |
| RASAT + PICARD (Qi et al., 2022) | 75.3 | 80.5 | - | - |
| RESDSQL(T5-3B)+NATSQL (Li et al., 2023) | 80.5 | 84.2 | - | - |
| **RESDSQL(T5-3B)+NATSQL+ReFSQL** | **83.1** | **86.2** | **-** | **-** |
| RESDSQL(Flan-T5)+NATSQL | 84.8 | 87.0 | 90.2 | 92.8 |
| **RESDSQL(Flan-T5)+NATSQL+ReFSQL** | **86.6** | **88.1** | **91.7** | **93.4** |

the questions. In contrast, Spider-Syn comprises 1034 samples where synonyms are used to replace schema-related words in the questions. Lastly, Spider-Realistic, comprising 508 samples, involves the removal of explicitly mentioned column names in the questions to simulate real-world scenarios with more ambiguous queries.

**WikiSQL** (Zhong et al., 2017) is a typical table-based question answering dataset, which has 80654 hand-annotated examples of questions, SQL queries and the corresponding answers from execution.

### 4.2 Baseline Models and Evaluation Metrics

We compare the proposed model with several baseline methods, including the current state-of-the-art model over the two benchmark datasets.

- **T5:** (Shaw et al., 2020) applies the pre-trained T5 to text-to-SQL task.

- **RATSQL:** (Qi et al., 2022) improves the encoder by adding the relation-aware self-attention module.

- **RESDSQL:** (Li et al., 2023) the schema is ranked and relevant schema items are injected into the encoder, while the decoder generates the skeleton first and then the SQL query.

- **PICARD:** (Scholak et al., 2021) improves the decoder by constraining beam search to generate grammatically correct SQL queries.

- **RAT-SQL + GRAPPA** (Yu et al., 2020) designs a schema item classification pre-training

task to adapt the seq-to-seq model to the structured input.

- **LGESQL** (Cao et al., 2021) integrates relational features using a line graph, and introduces an auxiliary task, called graph pruning, to enhance the encoder's capability.

- **UNIFIEDSKG** (Xie et al., 2022) unifies 21 structured knowledge grounding tasks into one single text-to-text model.

- **Uni-Parser** (Liu et al., 2022) proposes a unified semantic parser method for question answering (QA) on both KB and DB.

- **UNISAR** (Dou et al., 2022) proposes a unified structure-aware model to solve text-to-SQL across various settings.

- **ChatGPT** (Liu et al., 2023) explores using chatgpt to solve text-to-SQL tasks.

To assess the performance of the Text-to-SQL parser, we employ three evaluation metrics: Exact-set-Match accuracy (EM), Execution accuracy (EX), and answer accuracy (F1) (Liu et al., 2022; Zhong et al., 2017). The EM metric determines whether the predicted SQL query can precisely match the gold SQL query by converting them into a specialized data structure. On the other hand, the EX metric compares the execution results of the predicted SQL query with the gold SQL query. In practice, we combine the EM and EX scores to evaluate the model and select the model with the best overall performance.

### 4.3 Results on Spider and WikiSQL

We implement comprehensive experiments to verify the effectiveness of our framework. The framework is integrated with the RESDSQL backend model, which utilizes fine-tuned techniques. The experimental results are presented in Table 1. Our ReFSQL framework, when applied to the RESDSQL backbone model, outperforms all baseline models on the two benchmark datasets. The approach achieves state-of-the-art performance in methods that employ the fine-tuning approach. Specifically, after adapting to our framework, the RESDSQL-based model improves the EM by 1.8 on Spider and 1.5 on WiKiSQL compared with the original model. In addition, our framework also has an improvement effect on other metrics.

Table 2: Evaluation results of our framework adapted with different models(%).

| Approach | Spider | |
| --- | --- | --- |
| | DevEM. | DevEX. |
| T5-small | 47.6 | 47.8 |
| **T5-small+ReFSQL** | **54.3** | **53.9** |
| T5-Base | 57.2 | 57.9 |
| **T5-Base+ReFSQL** | **61.8** | **61.7** |
| T5-Large | 65.3 | 67.2 |
| **T5-Large+ReFSQL** | **67.8** | **69.8** |
| T5-3B | 71.5 | 74.4 |
| **T5-3B+ReFSQL** | **73.6** | **76.5** |
| RASAT+PICARD | 75.3 | 80.5 |
| **RASAT+PICARD+ReFSQL** | **77.2** | **82.1** |
| RESDSQL(T5-3B)+NATSQL | 80.5 | 84.2 |
| **RESDSQL(T5-3B)+NATSQL+ReFSQL** | **83.1** | **86.0** |
| RESDSQL(Flan-T5)+NATSQL | 84.8 | 87.0 |
| **RESDSQL(Flan-T5)+NATSQL+ReFSQL** | **86.6** | **88.1** |

### 4.4 Analyses on Different Models

As mentioned above, our framework can be used for different fine-tuning approaches. To further validate the performance of our framework, we have integrated our framework with some different models, the results are shown in Table 2.

Our framework demonstrates flexibility and effectiveness, allowing for adaptation with numerous backbone models. Remarkably, our framework yields notable improvements when applied to small-size models like T5-small. This suggests that our framework can bridge the performance gap between small and large models, achieving comparable effects. Furthermore, we conducted experiments using a larger scale model, Flan-T5, and observed a substantial improvement of nearly 2%. These results indicate that our framework can

deliver impressive performance even on large language models. Additionally, we investigated the impact of model size. As illustrated in Table 2, our framework consistently showcases larger performance gaps when compared to its T5-3B counterpart. This observation aligns with previous findings in other fine-tuning tasks, highlighting the ability of larger pre-trained models to capture more knowledge effectively.

Table 3: Evaluation results of removing different modules of our framework (%).

| Approach | Spider | |
| --- | --- | --- |
| | DevEM. | DevEX. |
| **RESDSQL(Flan-T5)+NATSQL+ReFSQL** | **86.6** | **88.1** |
| RESDSQL(Flan-T5)+NATSQL+ReFSQL(w/O the SR) | 85.3 | 87.4 |
| RESDSQL(Flan-T5)+NATSQL+ReFSQL(w/O the CLM) | 86.0 | 87.8 |

### 4.5 Ablation Study

We conducted a comprehensive ablation study on the development set of the Spider dataset to thoroughly examine the individual impact of each module within our framework.

- **Structure-enhanced retriever** To assess the efficacy of the structure-enhanced retriever, we conducted an experiment. Instead, we relied solely on Bert for calculating the semantic similarity between questions. We selected samples with higher ranks as positive examples and randomly sampled other samples as negative examples within the batch. The results, as presented in Table 3, demonstrate a decrease in EM and EX scores when this module is removed. This outcome suggests that the retriever plays a crucial role in obtaining similar samples and facilitating the model's acquisition of specific knowledge during the text-to-SQL process.

- **Mahalanobis Contrastive Learning mechanism** We further investigate the efficacy of the mahalanobis contrastive learning mechanism in our framework. The results, presented in Table 3, indicate that removing this contrastive learning mechanism results in a decrease in the EM and EX metrics. This finding suggests that the contrastive learning mechanism plays a crucial role in guiding the representation of samples toward a distribution that encompasses similar samples.

Table 4: Evaluation results on Spider-DK, Spider-Syn, and Spider-Realistic (%)..

| Approach | Spider-DK | | Spider-Syn | | Spider-Realistic | |
| --- | --- | --- | --- | --- | --- | --- |
| | EM. | EX. | EM. | EX. | EM. | EX. |
| LGESQL + ELECTRA (Xu et al., 2020) | 48.4 | - | 64.6 | - | 69.2 | - |
| T5-3B+PICARD (Shaw et al., 2020) | - | - | 59.4 | 65.3 | 63.2 | 65.0 |
| RASAT + PICARD (Qi et al., 2022) | - | - | - | - | 69.7 | 71.9 |
| RESDSQL(T5-3B)+NATSQL (Li et al., 2023) | 53.3 | 66.0 | 69.1 | 76.9 | 77.4 | 81.9 |
| RESDSQL(Flan-T5)+NATSQL | 55.6 | 67.6 | 71.3 | 79.2 | 79.6 | 84.2 |
| **RESDSQL(Flan-T5)+NATSQL+ReFSQL** | **56.9** | **69.2** | **72.8** | **80.6** | **82.4** | **86.5** |

## 4.6 Robustness

To evaluate the robustness of our framework, we train our model on the training set of the Spider dataset and assess its performance on three challenging Spider variants: Spider-DK, Spider-Syn, and SpiderRealistic. The results, presented in Table 4, reveal a surprising and significant performance advantage of RESDSQL(Flan-T5)+NatSQL+ReFSQL overall strong competitors across all three datasets. This finding suggests that our framework can also enhance the robustness of Text-to-SQL parsers.

## 5 Related Work

### 5.1 Text-to-SQL

In recent studies, significant advancements have been made in enhancing the performance of seq-to-seq models for text-to-SQL tasks, focusing on both the encoding and decoding stages. (Qi et al., 2022) introduce relation-aware self-attention into the T5 encoder, enabling the capture of important structural information such as schema linking. (Scholak et al., 2021) proposes a beam search constraint during inference to ensure the generation of grammatically correct decoding results. (Li et al., 2023)enhance the encoder by incorporating the most relevant schema items into the input sequence, leading to improved performance.

### 5.2 Retrieval-augmented generation

Retrieval-augmented generation, which incorporates large language models with external retrieval modules, has achieved impressive performance in various tasks in recent years (Li et al., 2022a). One line of research focuses on enhancing language models with retrieval modules to provide additional knowledge (Si et al., 2022; Borgeaud et al., 2022). Another approach involves leveraging retrieval techniques to extract useful information from the training data. For example, (Long et al., 2022) fuse the base image encoder with relevant images retrieved from the training data to address the challenge of long-tail visual recognition. (Xiao et al., 2021) incorporate relevant sentences in the target style to improve the unsupervised style transfer model

## 6 Conclusion

This paper presents a research focus on the Text-to-SQL task, aiming to improve SQL generation through a retrieval-augmented framework called Ref-SQL. The main objective is to address the gap between specific knowledge and general knowledge. To obtain specific knowledge, a structure-enhanced retriever is devised to identify similar samples based on question semantics and schema structure. Additionally, a contrastive learning approach is employed to facilitate the transfer of samples towards a similar semantic distribution, further mitigating the aforementioned gap. The effectiveness of RefSQL is evaluated on five widely-used benchmark datasets, where it surpasses all baseline models, verifying its superior performance.

## 7 Limitations

To train the retrieval model effectively, it is necessary to allocate sufficient computing resources. Additionally, our research focus is limited to the English language due to its availability of a wide range of analytical tools and resources, which surpass those available for other languages.

## 8 Acknowledgement

Thanks to reviewers for their helpful comments on this paper. This paper is funded by the the National Natural Science Foundation of China (No.62172393, U1836206 and U21B2046), Zhongyuanyingcai program-funded to central plains science and technology innovation leading

talent program (No.204200510002), AntGroup Research Intern Program and Major Public Welfare Project of Henan Province (No.201300311200).

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

## A   Example Appendix

This is a section in the appendix.