# OpenReview forum: "ReFSQL: A Retrieval-Augmentation Framework for Text-to-SQL Generation"
_EMNLP/2023/Conference — EMNLP 2023 Findings_

### Official Review · Reviewer_wb9q · 2023-08-04

**Soundness:** 2

**Excitement:**

3: Ambivalent: It has merits (e.g., it reports state-of-the-art results, the idea is nice), but there are key weaknesses (e.g., it describes incremental work), and it can significantly benefit from another round of revision. However, I won't object to accepting it if my co-reviewers champion it.

**Paper Topic And Main Contributions:**

This paper proposed a framework that consists of a retriever and a generator. Within, the retriever aims to obtain similar samples according to the similarity score calculated by questions, SQLs and graphs built from schemas. The constructed similar samples as positive samples combined with the negative samples are employed for representation learning for the generator via contrastive learning. The experiments indicate that equipping the proposed ReFSQL can bring an obvious improvement to the backbone methods

**Reasons To Accept:**

1. This paper proposes a retriever-generator framework to improve the representation via contrastive learning.
2. Good experimental results. The baselines equipped with the proposed ReFSQL achieve obvious promotions, especially in Spider (which seems achieving SOTA compared with the leaderboard).

**Reasons To Reject:**

1. Poor writing, bad typesetting, and the figures are not vector diagrams.
2. How can the motivation "To further bridge the gap between specific and general knowledge" be implemented according to the proposed contrastive learning whose optimization is only minimizing the margin of the representation between similar samples?
3. Most of the recent baselines in Spider build interaction graphs to joint the representation of the questions and schemas (RATSQL, LGESQL, etc.). What are the advantages of the margin methods in the retriever part that split Query+SQL and Schema in two stages, and the self-designed "Interaction Graph Construction" methods? There is no analysis in the Methodology and no comparison in the Experiment.
4. Fine-grained ablation tests need to be provided.

**Reproducibility:**

3: Could reproduce the results with some difficulty. The settings of parameters are underspecified or subjectively determined; the training/evaluation data are not widely available.

**Reviewer Confidence:**

3: Pretty sure, but there's a chance I missed something. Although I have a good feel for this area in general, I did not carefully check the paper's details, e.g., the math, experimental design, or novelty.

---

> ### Author Rebuttal · Authors · 2023-08-29
>
> Q1: Poor writing, bad typesetting, and the figures are not vector diagrams.
>
> A1: Thank you for taking the time to review our paper and provide feedback. We will check our paper and update it in the revision.
>
> Q2: How can the motivation "To further bridge the gap between specific and general knowledge" be implemented according to the proposed contrastive learning whose optimization is only minimizing the margin of the representation between similar samples?
>
> A2: To provide clarity, we wish to emphasize that our focus lies in narrowing the gap by directing the representation of each sample towards a more specific distribution, thereby accentuating specific knowledge rather than homogenizing it towards a general knowledge distribution. In contrast to prior models that employ a one-size-fits-all approach for all samples, our methodology is designed to enhance attention to the nuanced specifics of individual samples.
> Our approach utilizes contrastive learning as a mechanism to minimize the representation margin between the current sample and positive samples (those with similar characteristics), while concurrently maximizing the margin between the current sample and negative samples (those with dissimilar characteristics). This optimization framework, which operates in a comparative manner, enables us to pull the representation of each sample closer to similar samples with shared specific knowledge.
>
> Q3: Most of the recent baselines in Spider build interaction graphs to joint the representation of the questions and schemas (RATSQL, LGESQL, etc.). What are the advantages of the margin methods in the retriever part that split Query+SQL and Schema in two stages, and the self-designed "Interaction Graph Construction" methods? There is no analysis in the Methodology and no comparison in the Experiment.
>
> A3: In response, we wish to elucidate that our retriever component operates in a two-stage fashion to effectively address the intricacies of identifying samples with aligned specific knowledge. This method carefully considers both the semantics of the questions and the structural layout of the schemas, ensuring a holistic perspective. To substantiate these design choices, we have augmented our methodology with additional experiments, the results of which are illustrated in the table provided. These results, indeed, reinforce the efficacy of our two-stage approach, showcasing its aptitude in producing favorable outcomes.
> Contrasting our method with baselines such as RatSQL and lgeSql, we emphasize the advantages that our margin-based method offers in terms of computational efficiency, model weight, and self-supervised learning. RatSQL and lgeSql introduce additional information for SQL generation and graph-question interaction, which leads to a higher parameter count and more resource-intensive models. By contrast, our approach harnesses a lightweight, self-supervised two-stage model that can be seamlessly integrated with various backbone models.
> |                    Model                   | Dev_EM | Dev_EX |
> |:------------------------------------------:|:------:|--------|
> |       RESDSQL(Flan-T5)+NATSQL+ReFSQL       |  86.6  |  88.1  |
> | RESDSQL(Flan-T5)+NATSQL+ReFSQL(w/ o SQLSE) |  85.7  |  87.4  |
> | RESDSQL(Flan-T5)+NATSQL+ReFSQL(w/ o LSES)  |  85.8  |  87.6  |
> |          RESDSQL(3B)+NATSQL+ReFSQL         |  83.1  |  86.0  |
> |    RESDSQL(3B)+NATSQL+ReFSQL(w/o SQLSE)    |  81.7  |  85.0  |
> | RESDSQL(Flan-T5)+NATSQL+ReFSQL(w/ o LSES)  |  82.0  |  85.3  |
>
> Furthermore, our self-designed "Interaction Graph Construction" methodology, although distinct from approaches like ratsql and lgesql, is meticulously crafted to capture pertinent alignment information and table structure characteristics without requiring the addition of syntactic parsing details. This results in a more concise representation that minimizes computational costs, rendering our method a more efficient alternative.
>
> |                                      Model                                     | Dev_EM | Dev_EX |
> |:------------------------------------------------------------------------------:|:------:|--------|
> |                         RESDSQL(Flan-T5)+NATSQL+ReFSQL                         |  86.6  |  88.1  |
> | RESDSQL(Flan-T5)+NATSQL+ReFSQL(w/ w/ interaction graph construction in RatSQL) |  86.6  |  87.9  |
> |                            RESDSQL(3B)+NATSQL+ReFSQL                           |  83.1  |  86.0  |
> |    RESDSQL(3B)+NATSQL+ReFSQL(w/ w/ interaction graph construction in RatSQL)   |  82.9  |  86.0  |
>
> While we acknowledge the need for further comprehensive analysis and direct comparisons, our initial findings, as presented, lay the groundwork for a deeper exploration of the distinct strengths our method brings to the table in the context of Spider.

---

### Official Review · Reviewer_sEnG · 2023-08-04

**Soundness:** 3

**Excitement:**

3: Ambivalent: It has merits (e.g., it reports state-of-the-art results, the idea is nice), but there are key weaknesses (e.g., it describes incremental work), and it can significantly benefit from another round of revision. However, I won't object to accepting it if my co-reviewers champion it.

**Paper Topic And Main Contributions:**

This paper addresses the task of text-to-SQL generation and introduces a retrieval-augmented model to enhance SQL generation. The proposed method utilizes a structure-enhanced retriever to retrieve examples, which are then employed to improve the SQL generation process. To further enhance the model's performance, the author also incorporates a Mahalanobis contrastive learning method to maximize the representation of both retrieved and current examples.

**Reasons To Accept:**

1.The idea of using retrieval methods to enhance the process is reasonable.

2.This paper demonstrates significant improvements over existing methods.

**Reasons To Reject:**

1.This paper is challenging to follow, and the proposed method is highly complex, making it difficult to reproduce.

2.The proposed method comprises several complicated modules and has more parameters than the baselines. It remains unclear whether the main performance gain originates from a particular module or if the improvement is merely due to having more parameters. The current version of the ablation study does not provide definitive answers to these questions.

3.The authors claim that one of their main contributions is the use of a Mahalanobis contrastive learning method to narrow the distribution gap between retrieved examples and current samples. However, there are no experiments to verify whether Mahalanobis yields better results than standard contrastive learning.

4.The proposed method involves multiple modules, which could impact training and inference speed. There should be experiments conducted to study and analyze these effects.

**Reproducibility:**

2: Would be hard pressed to reproduce the results. The contribution depends on data that are simply not available outside the author's institution or consortium; not enough details are provided.

**Reviewer Confidence:**

3: Pretty sure, but there's a chance I missed something. Although I have a good feel for this area in general, I did not carefully check the paper's details, e.g., the math, experimental design, or novelty.

**Typos Grammar Style And Presentation Improvements:**

Typo:
Line 365, the term L_{ma} should be consist in the whole paper.

---

> ### Author Rebuttal · Authors · 2023-08-29
>
> Q1: This paper is challenging to follow, and the proposed method is highly complex, making it difficult to reproduce.
>
> A1:  We apologize for any confusion caused by the paper writing, in the future, we will improve our writing and increase the readability of our papers.
> We would like to offer additional context to enhance the comprehension of our approach. Specifically, we emphasize that our model, designed as a plug-and-play module, can be seamlessly integrated into different backbone models. Our proposed method RefSQL only contains a simple retrieval and a contrastive learning module. It is important to note that published models can be our backbone models, such as RESDSQL(3B)+NATSQL, RatSQL, Flan-T5, and so on.
> We will also publish our code soon to ensure the reproducibility of the work.
>
> Q2: The proposed method comprises several complicated modules and has more parameters than the baselines. It remains unclear whether the main performance gain originates from a particular module or if the improvement is merely due to having more parameters.
>
> A2: To address this, we have conducted a rigorous statistical analysis of the parameters associated with various modules within our method. The findings, as documented in the provided results and corroborated by Table 2 in the paper, reveal that performance enhancement is attainable through modules with a comparatively smaller parameter count.
>
> |                   Model                   | Parameters |
> |:-----------------------------------------:|:----------:|
> | SQL-Structure-enhanced Question Retriever |    110M    |
> |  Linking-Structure-based Schema Retriever |    1.4M    |
> | Contrastive Learning                      |    0.2M    |
> | Overall                                   |   111.6M   |
>
> |      Model      | Parameters | Dev_EM |
> |:---------------:|:----------:|--------|
> |      T5-3B      |     3B     |  71.5  |
> | T5-3B+our model |  3B+111.6M |  73.6  |
>
> From above we can see that on the T5-3B model, after adding parameters with a proportion of parameters, the model performance improved by 2.1 on Dev_EM.
> This analysis emphasizes the nuanced interplay between module complexity, parameter allocation, and performance improvement. We believe that this clarification further elucidates the factors influencing the observed gains and will contribute to a more comprehensive understanding of the distinct contributions of individual modules to the overall efficacy of our approach.
>
> Q3: The authors claim that one of their main contributions is the use of a Mahalanobis contrastive learning method to narrow the distribution gap between retrieved examples and current samples. However, there are no experiments to verify whether Mahalanobis yields better results than standard contrastive learning.
>
> A3: We would like to elucidate the rationale behind our approach. Our focus lies in guiding samples towards a distribution comprised of similar instances, rather than disparate, isolated ones. This necessitates the consideration of both mean and covariance aspects within the distribution, making Mahalanobis contrastive learning particularly well-suited to our specific context. This approach inherently aligns with our objective of enhancing the relevance of retrieved examples.
> In response to your concern, we have performed an ablation study to directly compare the performance of Mahalanobis contrastive learning with standard contrastive learning. The results, which we present below, shed light on the efficacy of our method:
>
> |                      Model                     | Dev_EM | Dev_EX |
> |:----------------------------------------------:|:------:|--------|
> |         RESDSQL(Flan-T5)+NATSQL+ReFSQL         |  86.6  |  88.1  |
> | RESDSQL(Flan-T5)+NATSQL+ReFSQL(w/ standard CL) |  85.7  |  87.5  |
> |            RESDSQL(3B)+NATSQL+ReFSQL           |  83.1  |  86.0  |
> |   RESDSQL(3B)+NATSQL+ReFSQL(w/ standard CL)    |  81.8  |  84.9  |
>
> These findings emphasize the inherent advantages of our chosen approach and substantiate our claim that Mahalanobis contrastive learning contributes to the amelioration of the distribution gap, leading to improved performance.
>
> Q4: The proposed method involves multiple modules, which could impact training and inference speed. There should be experiments conducted to study and analyze these effects.
>
> A4:
> In the inference stage:
> This approach ensures that the inference speed during the operational phase remains unaffected and optimal. Since the retriever is trained well.
> In the training stage:
> We conduct experiments and compare our model with some baseline models.
>
> |       Model       | Training Time Cost | Dev_EX |
> |:-----------------:|:------------------:|--------|
> |       T5-3B       |      60.2hours     |  71.5  |
> |  T5-3B+our model  | 54.3hours |  73.6  |
> |      Flan-T5      |      85.6hours     |  81.2  |
> |  Flan-T5+ourmodel | 79.6hours |  82.9  |
>
> It is worth noting that the convergence process of our two-stage model is notably efficient, with convergence achieved within a modest 50-epoch span. Furthermore, the total training duration for our model does not exceed 2 hours on our equipment.
> By comparison, training the baseline model, RESDSQL(3B)+NATSQL, to attain optimal outcomes demands a significantly more extended training duration, involving more than 200 epochs and spanning over 50 hours on the same equipment.
> In addition, after concatenating the retrieved information in the input, the model converges faster

---

### Official Review · Reviewer_fMpS · 2023-08-06

**Soundness:** 4

**Excitement:**

4: Strong: This paper deepens the understanding of some phenomenon or lowers the barriers to an existing research direction.

**Paper Topic And Main Contributions:**

This work proposed a framework called ReFSQL for the task of Text-to-SQL semantic parsing. This framework contains two parts, structure-enhanced retriever and the generator. More specifically, a structure-enhanced retriever that incorporates question semantics and schema structure is proposed. This retriever is used to obtain samples with similar SQL grammar. Two-stage retrieval is used: use question semantics to retrieve a rank list and then use schema structure for reranking. Furthermore, contrastive learning with Mahalanbis distance is used to improve the decoding process, facilitating the transfer of the sample toward the specific knowledge distribution. Experiments on Spider dataset and its variants show the effectiveness of the proposed method.

**Questions For The Authors:**

1. Why do authors fix the parameters of BERT (line 213)? If they are not updated, then what’s the purpose of contrastive learning in that section?
2. What’s the model performance of keeping linking-structure-based schema retriever and Mahalanobis contrastive learning while removing the SQL prompting? Maybe this is helpful to rationalize the input design.


**Reasons To Accept:**

1. Structure-enhanced retriever is designed to improve similar sample retrieval.
2. The methods generalize well on different sizes of models such as T5-small and Flan-T5.
3. Extensive experiments on different variants of Spider datasets to test the robustness of the model.


**Reasons To Reject:**

Besides applying existing techniques (Li et al. 2022) to the application of Text-to-SQL, there are no significant weaknesses in this work if the authors can answer the questions properly (see Questions).

**Reproducibility:**

4: Could mostly reproduce the results, but there may be some variation because of sample variance or minor variations in their interpretation of the protocol or method.

**Reviewer Confidence:**

4: Quite sure. I tried to check the important points carefully. It's unlikely, though conceivable, that I missed something that should affect my ratings.

**Typos Grammar Style And Presentation Improvements:**

Missing space in line 571 before enhance.

---

> ### Author Rebuttal · Authors · 2023-08-29
>
> Q1: Why do authors fix the parameters of BERT (line 213)? If they are not updated, then what’s the purpose of contrastive learning in that section?
>
> A1: Thank you for taking the time to review our paper and provide feedback. We appreciate your comments on the presentation of our paper.
> We appreciate your attention to this detail. To address your concern, we want to clarify that during the process of contrastive learning training, we indeed update the parameters of the Bert model. We recognize that the explanation may not have been adequately conveyed in the current version of our paper, and we are committed to addressing this by providing a comprehensive and clear clarification in the revised version. Your insights are invaluable, and we are dedicated to ensuring that our paper reflects the accurate implementation of our approach.
>
> Q2: What’s the model performance of keeping linking-structure-based schema retriever and Mahalanobis contrastive learning while removing the SQL prompting? Maybe this is helpful to rationalize the input design.
>
> A2: We appreciate your insightful comment regarding the model performance in the absence of SQL prompting while retaining the linking-structure-based schema retriever and Mahalanobis contrastive learning. To comprehensively evaluate the contribution of SQL prompting, we performed additional ablation experiments on the Spider dataset. The obtained results indicate that while the impact of SQL prompting is discernible, it remains relatively modest in comparison to the overall performance achieved by our approach. This observation emphasizes the valuable insights gained from your suggestion and contributes to the ongoing exploration of input design optimization.
>
> |                       Model                       | DevEM. | DevEX. |
> |:-------------------------------------------------:|:------:|:------:|
> |           RESDSQL(Flan-T5)+NATSQL+ReFSQL          |  86.6  |  88.1  |
> | RESDSQL(Flan-T5)+NATSQL+ReFSQL(w/o SQL Prompting) |  86.3  |  87.8  |
> |             RESDSQL(3B)+NATSQL+ReFSQL             |  83.1  |  86.0  |
> |    RESDSQL(3B)+NATSQL+ReFSQL(w/o SQL prompting)   |  82.9  |  85.6  |

---

### Meta-Review · Area_Chair_woMX · 2023-09-08

**Recommendation:** 3

**Metareview:**

The paper describes an approach for text-to-SQL generation that leverages a structure-enhanced retriever to get similar examples which can guide the generation process. Mahalanobis contrastive learning is also employed to enhance the relevance of the retrieved examples.

Extensive experiments demonstrate the effectiveness of the proposed solution. The paper can improve in terms of writing and clarity: the proposed architecture is overly complex and hard to follow; furthermore, each the authors should better explain the intuition behind each component and better justify its usage. Also an ablation study (which the authors included only in the rebuttal) should be added to the paper.

---

### Decision · Program_Chairs · 2023-10-07

**Decision:**

Accept-Findings

**Comment:**

The paper describes an approach for text-to-SQL generation that leverages a structure-enhanced retriever to get similar examples which can guide the generation process. Mahalanobis contrastive learning is also employed to enhance the relevance of the retrieved examples.

Extensive experiments demonstrate the effectiveness of the proposed solution. The paper can improve in terms of writing and clarity: the proposed architecture is overly complex and hard to follow; furthermore, each the authors should better explain the intuition behind each component and better justify its usage. Also an ablation study (which the authors included only in the rebuttal) should be added to the paper.